# Structural Reorganization of Imidazolium Ionic Liquids Induced by Pressure-Enhanced Ionic Liquid—Polyethylene Oxide Interactions

**DOI:** 10.3390/ijms22020981

**Published:** 2021-01-19

**Authors:** Teng-Hui Wang, Li-Wen Hsu, Hai-Chou Chang

**Affiliations:** Department of Chemistry, National Dong Hwa University, Shoufeng, Hualien 974, Taiwan; 810712101@gms.ndhu.edu.tw (T.-H.W.); 610712001@gms.ndhu.edu.tw (L.-W.H.)

**Keywords:** polyethylene oxide (PEO), ionic liquids (ILs), high-pressure infrared spectroscopy

## Abstract

Mixtures of polyethylene oxide (PEO, M.W.~900,000) and imidazolium ionic liquids (ILs) are studied using high-pressure Fourier-transform infrared spectroscopy. At ambient pressure, the spectral features in the C–H stretching region reveal that PEO can disturb the local structures of the imidazolium rings of [BMIM]^+^ and [HMIM]^+^. The pressure-induced phase transition of pure 1-butyl-3-methylimidazolium bromide ([BMIM]Br) is observed at a pressure of 0.4 GPa. Pressure-enhanced [BMIM]Br-PEO interactions may assist PEO in dividing [BMIM]Br clusters to hinder the aggregation of [BMIM]Br under high pressures. The C–H absorptions of pure 1-hexyl-3-methylimidazolium bromide [HMIM]Br do not show band narrowing under high pressures, as observed for pure [BMIM]Br. The band narrowing of C–H peaks is observed at 1.5 GPa for the [HMIM]Br-PEO mixture containing 80 wt% of [HMIM]Br. The presence of PEO may reorganize [HMIM]Br clusters into a semi-crystalline network under high pressures. The differences in aggregation states for ambient-pressure phase and high-pressure phase may suggest the potential of [HMIM]Br-PEO (M.W.~900,000) for serving as optical or electronic switches.

## 1. Introduction

Polyethylene oxide (PEO) with electron donor groups (such as ethers) is well-known as a long-lasting host polymer for solid polymer electrolytes (SPEs) for energy storage devices [1,2,3]. PEO shows excellent properties such as a flexible backbone, high dielectric constant, good alkali metal cation association, low glass transition temperature, and high melting point for applications in SPE for energy storage devices [1,2,3,4]. SPE containing PEO mixed with low-lattice-energy salts can be used in devices such as lithium ion batteries, owing to the specific coordination of PEO with alkali metal ions [3]. PEO is a semi-crystalline polymer composed of ethylene oxide units with the general structural formula, H(OC_2_H_4_)_n_OH. PEO mixed with various amounts of plasticizers can modify the rigidity by changing the crystallization ratio (percentage of crystalline phase/percentage of amorphous phase) to enhance the ion transport properties [1,2,3,5,6,7]. The crystallization ratio of PEO can be easily modified by adding solvents, salts, and ionic liquids (ILs), while the geometric structures of PEO mixtures are related to the interactions between PEO and additives. The variation in the crystallization ratio of PEO induced by the additives can modify specific properties such as flexibility, stability, and ionic conductivity [5,6,7,8,9,10].

ILs are emerging green solvents owing to their low evaporation and other unique properties [10,11,12,13,14,15,16]. The anomalous behaviors and structure-property relationship of ILs were reviewed by Esperanca et al. [11] and Silva et al. [12], respectively, in the recent. In general, the structures of ILs containing 1-alkyl-3-methylimidazolium cations can be easily designed for specific applications in industrial processes [10,11,13]. The cations and anions of the imidazolium-ILs attract each other, but ions in ILs do not undergo ready packing. Thus, the liquid phase of ILs is typically observed at room temperature. The macroscopic properties of the ILs mainly depend on the interionic interactions of the cations and anions [17,18]. The presence of nano-segregation domains in ILs, i.e., heterogeneous microstructures, was reported in the literature [12]. The local nanoscale organization of imidazolium ILs can be described as supramolecular structures of ion pairs (cation-anion), clusters (cation-anion-cation-anion…), and/or three-dimensional network (cluster-cluster associations). The alkyl-length-dependent properties of 1-alkyl-3-methylimidazoliums have been intensively investigated in recent years [17,19,20,21,22]. Varying the alkyl side-chain length of 1-alkyl-3-methylimidazoliums can modify the association, aggregation, and three-dimensional network of the imidazoliums and anions in the ILs, and the density, viscosity, ion diffusivity, ionic strength, and electrochemical properties are also affected [8,9,23,24,25,26].

The linear polymer PEO is generally water-soluble owing to the presence of the ethylene oxide group, and the oxygen atoms of PEO are promising candidates as hydrogen-bond acceptors (HBAs) [20,27,28]. Researchers have shown that some associations between IL cations and anions may be substituted by the interactions of IL cations and PEO oxygen atoms upon mixing PEO with ILs [27,28]. Using molecular dynamics simulations, it has been determined that the distances between the imidazolium rings of ILs and oxygen atoms of PEO are slightly shorter than those of the cations and anions in the IL polymer electrolytes [4,20]. Imidazolium cations and PEO polymer appear to show non-negligible associations in the SPE systems [9,17,19]. To determine the pressure-dependent characteristics of the IL-PEO matrix systems, ILs with a simple spherical anion (Br^−^) and two alkyl chain lengths (butyl and hexyl) of 1-alkyl-3-methylimidazolium cations were selected for mixing with PEO to afford the IL/PEO mixtures in this study.

High-pressure IR spectroscopy is a direct method to investigate the pressure-induced changes in non-covalent interactions and intermolecular (or interionic) distances [29,30,31,32,33,34]. The local structures of the ILs can be disturbed by the polymers (PVdF-co-HFP [29], DNA [30]), β-cyclodextrin [31], porous silica [32], mica [33], and nano-alumina [33,34], as determined by the high-pressure method. For example, high-pressure studies indicate that the presence of PVdF-co-HFP mainly affects the local structures of the imidazolium C–H groups instead of the anionic groups [29]. Pressure-enhanced IL-polymer interactions are also observed [29,30]. In general, a variation in temperature may result in changes in both thermal energy and volume. However, the use of high-pressure methods allows the tuning of the intermolecular (or interionic) distances without changing the thermal energy. Herein, it is demonstrated that the length of the alkyl C–H side chain in ILs can significantly affect the pressure-induced phase transition in IL-PEO systems.

## 2. Results and Discussion

### 2.1. IR Study of 1-butyl-3-methylimidazolium Bromide ([BMIM]Br)-PEO Systems

Figure 1 shows the IR spectra of (a) pure 1-butyl-3-methylimidazolium bromide ([BMIM]Br), (b) PEO (M.W.~900,000) with 75 wt% [BMIM]Br, and (c) pure PEO (M.W.~900,000) at ambient pressure. The absorption profile of pure [BMIM]Br (Figure 1a) exhibits five major bands at 3138, 3067, 2958, 2934, and 2869 cm^−1^, corresponding to two imidazolium C–H bands (C^4,5^–H and C^2^–H) and three cation alkyl C–H peaks (methyl and butyl groups) [29,30,31,32,33,34]. Figure 1c shows the IR spectrum of pure PEO exhibiting multiple C–H stretching absorptions in the 2950–2850 cm^−1^ region [7,35]. As shown in Figure 1b, the absorption peaks at 2800–3000 cm^−1^ are composed of two components attributable to the alkyl C–H bands of [BMIM]^+^ and C–H stretching bands of PEO. Although the alkyl C–H absorption features show the overlapping contributions of [BMIM]^+^ and PEO, the absorptions at 3147 and 3088 cm^−1^ in Figure 1b are distinctly assigned to the blue-shifted imidazolium C^4,5^–H and C^2^–H, respectively. The blue-shifted imidazolium C–H bands in Figure 1b in comparison to those of pure [BMIM]Br in Figure 1a suggest that PEO may disturb the local structure of the imidazolium ring of [BMIM]^+^. Notably, imidazolium C^4,5^–H and C^2^–H show blue shifts of 9 and 21 cm^−1^, respectively, as shown in Figure 1b. Thus, the imidazolium C^2^–H group may be a favorable site for the IL-PEO interactions. Based on the results of ^1^H NMR and MD simulations [12], C^2^-H is most prone to the effects of the hydrogen bonding. For example, the relative distance of anion is close to C^2^-H and water for [BMIM][BF_4_]-water mixtures [12].

To analyze the experimental results in detail, the concentration dependence of (a) imidazolium C^4,5^–H and (b) imidazolium C^2^–H stretching bands of [BMIM]Br-PEO mixtures as a function of the weight percentage of [BMIM]Br is shown in Figure 2. In Figure 2b, the imidazolium C^2^–H shows noticeable blue shifts upon dilution with PEO in the concentration range of 100–75 wt% [BMIM]Br; mild shifts are observed at 75–10 wt% (Figure 2b). The blue shifts shown in Figure 2b in the 100–75 wt% region may be attributed to the interactions between the acidic moiety of the cation, that is, C^2^–H, and the hydrogen bond acceptor (HBA) of the polymer, that is, the ether groups. The concentration dependence of the C^4,5^–H band shown in Figure 2a reveals a similar tendency to that of C^2^–H, as shown in Figure 2b. Nevertheless, less blue shifts are observed for C^4,5^–H in Figure 2a due to a weaker acidity of the C^4,5^–H moiety in comparison to that of C^2^–H.

High pressures were utilized to characterize the interionic interactions of [BMIM]Br (Figure 3). Figure 3 shows the IR spectra of pure [BMIM]Br at (a) ambient pressure and (b) 0.4, (c) 0.7, (d) 1.1, (e) 1.5, (f) 1.8, and (g) 2.5 GPa. The C^4,5^–H and C^2^–H absorptions of pure [BMIM]Br at 3025–3200 cm^−1^ are split into three bands at 3130, 3100, and 3085 cm^−1^ with an increase in pressure to 0.4 GPa (Figure 3b). The three C^4,5^–H and C^2^–H peaks reveal mild shifts upon further compression from 0.4 to 2.5 GPa (Figure 3c–g). The narrowing and splitting of the imidazolium C–H bands may indicate the local structure organization of cations via pressure-enhanced interactions, as shown in Figure 3b. The clusters of IL may rearrange to form organized structures (or phase-transition) at 0.4 GPa. Notably, the pressure-induced phase transition at 0.4 GPa can only be observed for dried [BMIM]Br (pre-heated to 155 °C to remove moisture) as the presence of water appears to significantly affect the overall phase behavior of [BMIM]Br [36]. Further elevation in pressure (0.4–2.5 GPa) shows minor structural relaxations (Figure 3b–g). As shown in Figure 3, the alkyl C–H bands exhibit frequency shifts, and the spectral features change upon compression.

The IR spectra of the [BMIM]Br-PEO mixture (containing 80 wt% [BMIM]Br) at (a) ambient pressure and (b) 0.4, (c) 0.7, (d) 1.1, (e) 1.5, (f) 1.8, and (g) 2.5 GPa are shown in Figure 4. As shown, the C^2^–H and C^4,5^–H absorptions of the IL-PEO mixture with 80 wt% of [BMIM]Br show blue shifts to 3087 and 3151 cm^−1^, respectively, upon compression to 2.5 GPa. The C^4,5^–H and C^2^–H spectral profiles of the mixture with 80 wt% [BMIM]Br under high pressures (Figure 4) are significantly different from those of pure [BMIM]Br (Figure 3). The absence of band narrowing and splitting in Figure 4 may be attributed to the pressure-enhanced IL-polymer interactions. PEO may further divide aggregated [BMIM]Br into various clusters of different sizes via the cation-PEO interactions under high pressure. In other words, the local structures of C^4,5^–H and C^2^–H in Figure 4 are different from those of pure [BMIM]Br in Figure 3 under high pressures.

The pressure-dependent band shifts of imidazolium C^4,5^–H and C^2^–H of pure [BMIM]Br and the mixture with 80 wt% [BMIM]Br are shown in Figure 5. As shown, the imidazolium C–H absorptions of pure [BMIM]Br are separated into three bands accompanied by drastic shifts in frequency with an increase in the pressure to 0.4 GPa; the imidazolium C–H bands then undergo mild shifts in frequency with an increase in pressure to 2.5 GPa. The C^2^–H and C^4,5^–H bands of the mixture with 80 wt% [BMIM]Br show subtle shifts with pressure (Figure 5). The distinction in band shifts for both pure [BMIM]Br and the mixture with 80 wt% [BMIM]Br in Figure 5 may be attributed to the difference in the local environments of the imidazolium C^4,5^–H and C^2^–H moieties under high pressures (*p* > 0.4 GPa) [20,22].

### 2.2. IR Study of 1-hexyl-3-methylimidazolium Bromide ([HMIM]Br)-PEO Systems

To determine the effect of the alkyl chain length on cation-PEO interactions, 1-hexyl-3-methylimidazolium bromide ([HMIM]Br)-PEO mixtures were prepared and characterized using IR spectroscopy. Figure 6 shows the IR spectra of (a) pure [HMIM]Br, (b) mixture with 75 wt% [HMIM]Br, and (c) pure PEO at ambient pressure. Figure 6a (pure [HMIM]Br) shows three alkyl C–H bands at 2954, 2931, and 2859 cm^−1^ as well as two imidazolium C–H absorptions at 3136 and 3058 cm^−1^, corresponding to C^4,5^–H and C^2^–H, respectively. The C^4,5^–H and C^2^–H absorptions blue shift to 3141 and 3074 cm^−1^, respectively, (Figure 6b) with the addition of PEO to [HMIM]Br. The extents of blue shift, that is, 5 and 16 cm^−1^, for C^4,5^–H and C^2^–H, respectively, (Figure 6b) are similar to those of [BMIM]Br (Figure 1b). Thus, the IR measurements at ambient pressure are not sufficiently sensitive to distinguish the difference in interactions between the [BMIM]^+^-PEO and [HMIM]^+^-PEO systems.

Figure 7 shows the concentration dependence of the imidazolium C^4,5^–H and C^2^–H stretching frequencies for the [HMIM]Br-PEO mixtures as a function of the weight percentage of [HMIM]Br. As shown in Figure 7b, the C^2^–H band exhibits noticeable blue shifts as [HMIM]Br is diluted by PEO in the high-concentration region (≥80 wt%), followed by mild frequency shifts in the relatively low concentration region (80–10 wt% of [HMIM]Br). In contrast to the C^2^–H bands in Figure 7b, the C^4,5^–H absorptions show a mild frequency shift as a function of the [HMIM]Br concentration in Figure 7a. These observations may be attributed to the lower acidity of C^4,5^–H in comparison to that of the C^2^–H group.

Figure 8 shows the IR spectra of pure [HMIM]Br at (a) ambient pressure and (b) 0.4, (c) 0.7, (d) 1.1, (e) 1.5, (f) 1.8, and (g) 2.5 GPa. As shown in Figure 8, the absorptions of C^4,5^–H, C^2^–H, and alkyl C–H show mild shifts in frequency and peak broadening with an increase in pressure from ambient to 2.5 GPa. Notably, the C^4,5^–H and C^2^–H groups of pure [HMIM]Br (Figure 8) do not show band narrowing under high pressures as observed for pure [BMIM]Br (Figure 3). Based on the results in Figure 3 and Figure 8, the alkyl C–H length appears to be key factor that affects the local structures of imidazolium-based ILs under high pressures. Pure [HMIM]Br may associate to form glassy structures under high pressures (Figure 8) instead of organized structures shown in Figure 3 (pure [BMIM]Br). Researchers have suggested that an increase in the alkyl chain length in the 1-alkyl-3-methylimidazolium bis(trifluoromethylsulfonyl) imide ILs may decrease the electrostatic attraction between the imidazoliums and anions [9,17,20,22,23]. Thus, the proportion of hydrophobic associations and van der Waals interactions between the side chains may increase for the long-alkyl-chain ILs [21,22,24,26]. The alkyl chain length may affect the interionic interaction, aggregation, and average cluster sizes of the ILs under high pressures.

Figure 9 shows the IR spectra of the [HMIM]Br-PEO mixture containing 80 wt% [HMIM]Br at (a) ambient pressure and (b) 0.4, (c) 0.7, (d) 1.1, (e) 1.5, (f) 1.8, and (g) 2.5 GPa. The C^2^–H and C^4,5^–H absorptions of the mixture with 80 wt% [HMIM]Br reveal slight blue shifts and band broadening upon compression in Figure 9a–d (≤1.1 GPa). Surprisingly, the C^4,5^–H and C^2^–H absorptions are separated into four bands at 3067, 3102, 3117, and 3152 cm^−1^ under a pressure of 1.5 GPa (Figure 9e), where the band narrowing of cationic alkyl C–H peaks is also observed. The band narrowing observed in Figure 9e–g may be related to the decrease in inhomogeneous broadening. The sharper structures revealed in Figure 9e–g is in part due to conformation drift and anisotropic environment in a solid structures. For example, water is transformed from liquid to high-pressure ice VI and VII at the pressure of ~1 and ~2 GPa, respectively [37]. The reduction of O-H bandwidths, being obvious for high-pressure ice VI and VII, arises from the change of geometric properties of the hydrogen-bond network [37]. The spectral changes observed in Figure 9e can be attributed to the interplay of the PEO-PEO, IL-IL, and PEO-IL interactions. It is likely that the pressure-enhanced cation-PEO interactions can act a key factor under high pressures. [HMIM]Br can form a semi-crystalline structure (high-ordered 3-D structures of ions) in the presence of PEO under high pressures, and pressure-induced hexyl C–H-PEO interactions may be non-negligible. In other words, PEO may reorganize the [HMIM]Br associations via pressure-enhanced [HMIM]Br-PEO interactions. Researchers have revealed that the long alkyl chain disrupts the symmetry of each side of the imidazolium ring, and the presences of anions on the side of the long alkyl C-H chain is less favored [4]. A population of PEO atoms may be located around the long hexyl chain of [HMIM]^+^ under high pressures. The studies of the 1-alkyl-3-methylimidazolium bromide ILs indicate that some hydrogen bonds between the cation and Br^-^ are replaced by hydrogen bonding between the imidazolium and oxygen atom of PEO upon the addition of PEO to the ILs [27,28]. Furthermore, it has been concluded that for the 1-alkyl-3-methylimidazolium ILs, PEO can solvate the large imidazolium moieties with less stringent local modification via the van der Waals interaction with the PEO chains [4]. The results in Figure 9e–g may be attributed to the pressure-enhance PEO-imidazolium interaction and PEO-hexyl C–H chain interaction leading to IL-IL reorganization. Pressures used in this study (0.1 MPa–2.5 GPa) mainly change intermolecular (or interionic) distances (closer contact) and affect conformations [38]. Actually, in order to change the electronic structure of molecules or ions, pressure > 30 GPa are required. The abrupt changes (or dicontinuities) in frequency-shifts and bandwidths upon compression may provide the clues for conformation transitions or pressure-induced phase transitions. For example, pressure-induced reversible unfolding of biomolecules has drawn the attention of researchers [38]. It is known that aggregates/clusters of [HMIM]Br and [BMIM]Br are likely to be similar under ambient pressure. The almost identical trends of concentration-dependence observed in Figure 2 ([BMIM]Br-PEO) and Figure 7 ([HMIM]Br-PEO) suggest that IR measurements obtained under ambient pressure are not a sensitive tool to distinguish the local structures between [BMIM]Br-PEO and [HMIM]Br-PEO. This study indicates that high pressures can serve as a useful method to probe the differences in [BMIM]Br-PEO and [HMIM]Br-PEO interactions along the surface of PEO matrix under high pressures. Previous studies also revealed that the associated structures of [C_4_MIM][PF_6_]/DNA are more stable than those of [C_3_MIM][PF_6_]/DNA under high pressures [30].

The IR spectra of the mixture with 75 wt% [HMIM]Br at (a) ambient pressure and (b) 0.4, (c) 0.7, (d) 1.1, (e) 1.5, (f) 1.8, and (g) 2.5 GPa are shown in Appendix A. The band narrowing occurs at 1.1 GPa in Appendix A for the mixture with 75 wt% [HMIM]Br in comparison to 1.5 GPa for the mixture with 80 wt% [HMIM]Br (Figure 9e). The semi-crystalline structures are observed at different pressures for various mass ratios of PEO. Appendix A shows the IR spectra of the mixture with 50 wt% [HMIM]Br at (a) ambient pressure and (b) 0.4, (c) 0.7, (d) 1.1, (e) 1.5, (f) 1.8, and (g) 2.5 GPa. The drastic band narrowing observed in Figure 9e and Appendix A is not observed in Appendix A at high pressures. The pressure-dependent frequency shifts of C^2^–H and C^4,5^–H of pure [HMIM]Br and the mixtures are shown in Appendix A.

### 2.3. Possible Mechanism and Applications

Figure 10 illustrates the possible structural reorganization of [HMIM]Br in PEO matrices under high pressures. Elevation of pressure appear to force [HMIM]Br clusters to form ordered aggregation states in the presence of PEO. In other words, high pressure can be applied to tune the relative weight of ionic liquid aggregation state in [HMIM]Br-PEO systems, and the physical properties such as refractive index and conductivity may be altered due to structural reorganization. The differences in aggregation states for ambient-pressure phase and high-pressure phase may suggest the potential of [HMIM]Br-PEO (M.W.~900,000) for serving as optical or electronic switches.

To check the effect of PEO with other molecular weights (M.W.) on the organization of [HMIM]Br, the pressure dependence of [HMIM]Br -polyethylene glycol 1500 (PEG 1500, i.e., PEO with low M.W.) mixture and [HMIM]Br -ethylene glycol (EG) mixture were studied and displayed in Appendix A, respectively, in the Appendix A. As shown in Appendix A, the band-narrowing of C–H absorptions is not detected for the [HMIM]Br-PEG 1500 (PEO with low M.W.) and [HMIM]Br-EG mixtures containing 80 wt% of [HMIM]Br under high pressures. In other words, the organization of [HMIM]Br may be sensitive the molecular weight of polyethylene oxide under high pressures. We note that the [HMIM]Br-PEG 1500 and [HMIM]Br-EG mixtures are in the state of liquid under ambient pressure. Nevertheless, [HMIM]Br-PEO (MW. 900,000) mixtures are gel-like under ambient pressure. By the way, overlaid spectra (Figure 1, Figure 3, Figure 4, Figure 6, Figure 8 and Figure 9) are displayed in Appendix A, respectively, for the purpose of comparison.

## 3. Materials and Methods

The IL-PEO polymer electrolytes were prepared using PEO (average MW ~900,000, Sigma-Aldrich, St. Louis, MO, USA), 1-butyl-3-methylimidazolium bromide ([BMIM]Br, ≥ 97%, Fluka, Morris Plains, NJ, USA), 1-hexyl-3-methylimidazolium bromide ([HMIM]Br, 99%, Sigma-Aldrich, St. Louis, MO, USA), and acetonitrile (ACN, 99.9%, Merck, Darmstadt, Germany). [HMIM]Br-PEO and [BMIM]Br-PEO mixtures with different weight ratios of ILs were prepared by adding suitable amounts of ACN as the solvent. The mixtures were then dissolved, affording transparent sol-like solutions. Thereafter, ACN was removed from the sol-like products by placing these in air at 25 °C for at least one day. The samples were further dried at 155 °C using a moisture analyzer (MS-70, A&D Company, Tokyo, Japan) before performing spectral measurements. Polyethylene glycol 1500 (PEG 1500) and ethylene glycol (>95%) were purchased from Alfa Aesar (Tewksbury, MA, USA) and Showa Chemical (Tokyo, Japan), respectively.

In the laboratory, a diamond anvil cell (DAC) equipped with two type-IIa diamonds with a diamond culet size of 0.6 mm was used to generate high pressures (up to 2 GPa). The IR spectra were obtained using a Fourier-transform infrared spectrophotometer (Spectrum RXI, Perkin-Elmer, Naperville, IL, USA) combined with a beam condenser. The beam condenser was utilized to enhance the intensity of the passed IR beam. To eliminate the absorption of the diamond anvils, the absorption spectra of the DAC were measured and subtracted from those of the samples. The sample holder was prepared from a 0.25-mm thick Inconel gasket with a hole of 0.3-mm diameter. To prevent the saturation of the IR bands, transparent CaF_2_ crystals were placed into the sample holder and compressed firmly prior to inserting the samples. The pressure calibration was performed following the Wong method [39,40]. A resolution of 4 cm^−1^ (data point resolution of 2 cm^−1^) and 1000 scans were selected for high-pressure IR measurements.

## 4. Conclusions

High-pressure IR spectroscopy is used to investigate the ionic association of various mass ratios of ILs ([BMIM]Br or [HMIM]Br) mixed with PEO. At ambient pressure, both [BMIM]Br-PEO and [HMIM]Br-PEO mixtures show local structural changes in ILs disturbed by the ethylene oxide groups (HBA) of PEO. Under high pressures, the studies of [BMIM]Br-PEO mixtures reveal that PEO may inhibit [BMIM]Br aggregation by dividing [BMIM]Br clusters into various clusters of smaller sizes. However, in the [HMIM]Br-PEO systems, the presence of PEO may allow [HMIM]Br to coordinate into a semi-crystalline network via pressure-enhanced interaction between PEO and the long alkyl moiety of IL ([HMIM]^+^) under high pressures. The semi-crystalline structures of the [HMIM]Br-PEO mixtures are observed at different pressures for various mass ratios of PEO.

## Figures and Tables

**Figure 1 ijms-22-00981-f001:**
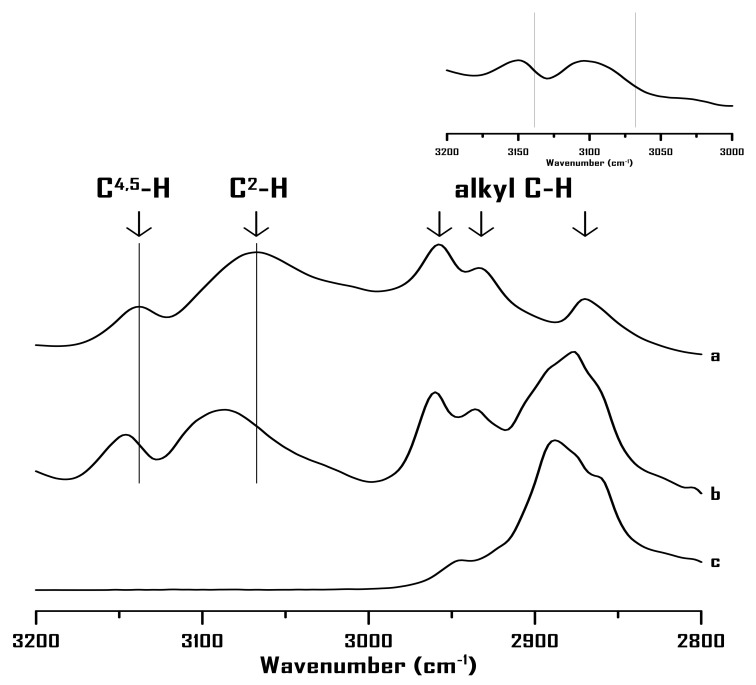
Infrared (IR) spectra of (**a**) pure [BMIM]Br, (**b**) mixture of PEO containing 75 wt% [BMIM]Br, and (**c**) pure PEO at ambient pressure. Upper-right corner: the difference spectrum (b-a).

**Figure 2 ijms-22-00981-f002:**
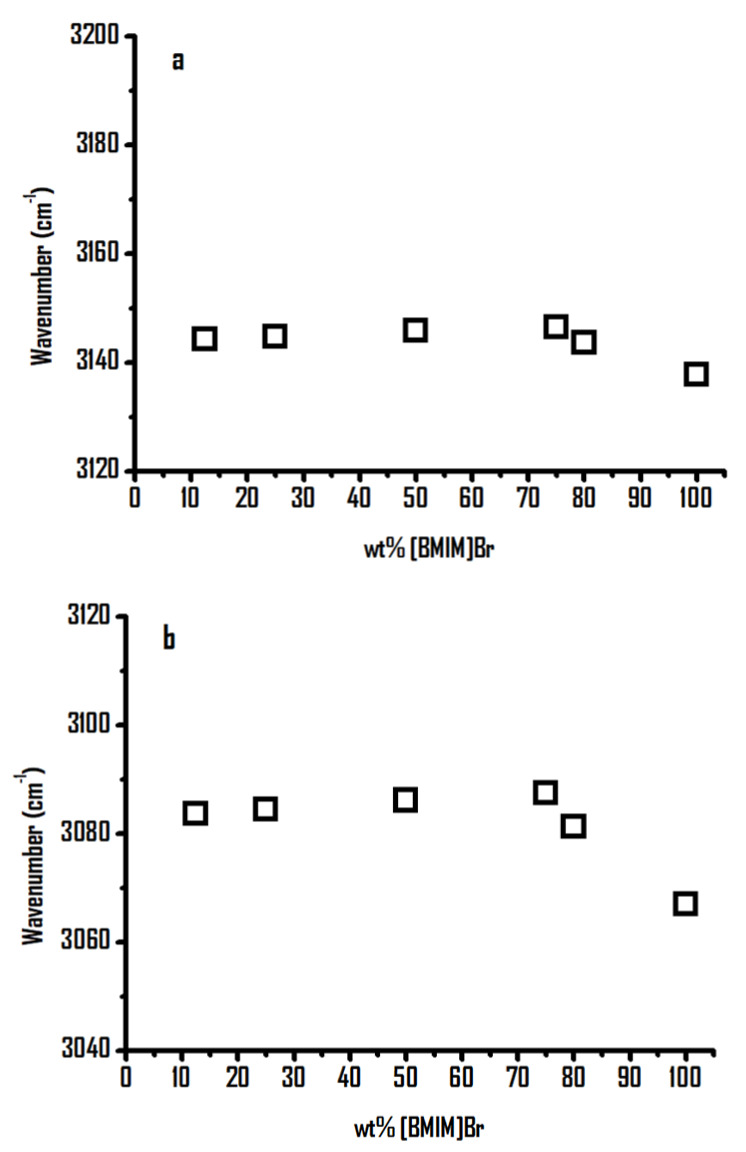
Concentration dependence of (**a**) imidazolium C^4,5^–H and (**b**) imidazolium C^2^–H stretching bands of [BMIM]Br-PEO mixtures as a function of the weight percentage of [BMIM]Br.

**Figure 3 ijms-22-00981-f003:**
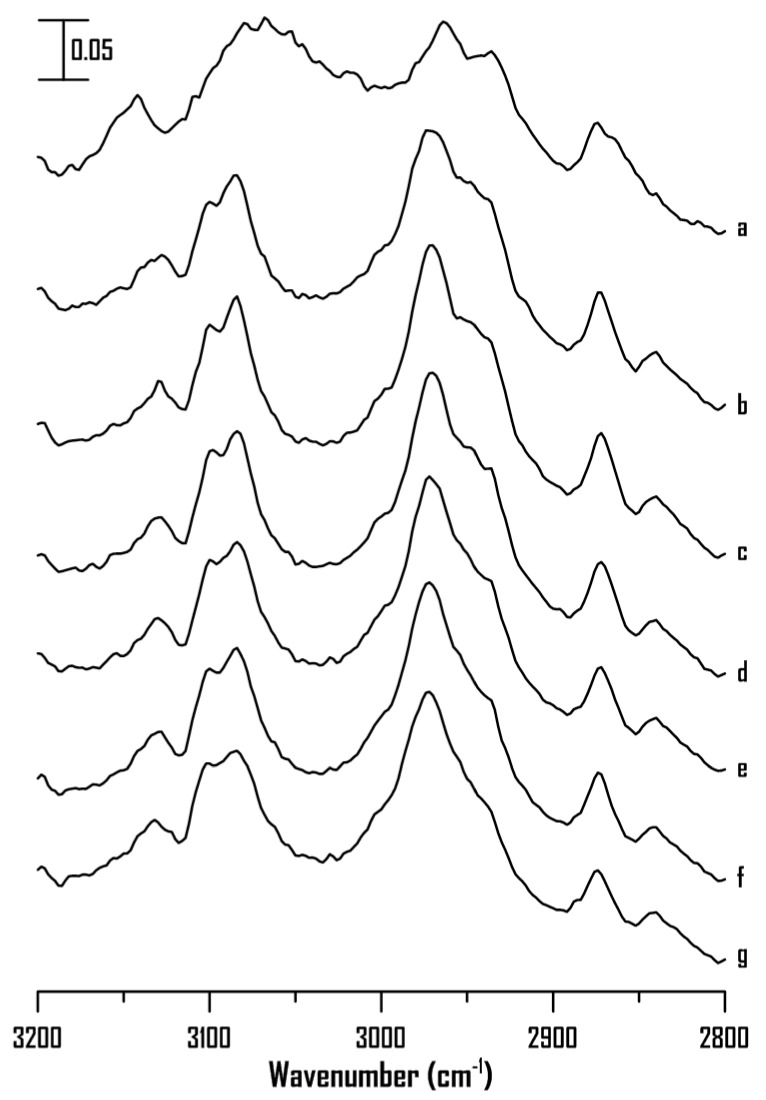
IR spectra of pure [BMIM]Br at (**a**) ambient pressure and (**b**) 0.4, (**c**) 0.7, (**d**) 1.1, (**e**) 1.5, (**f**) 1.8, and (**g**) 2.5 GPa.

**Figure 4 ijms-22-00981-f004:**
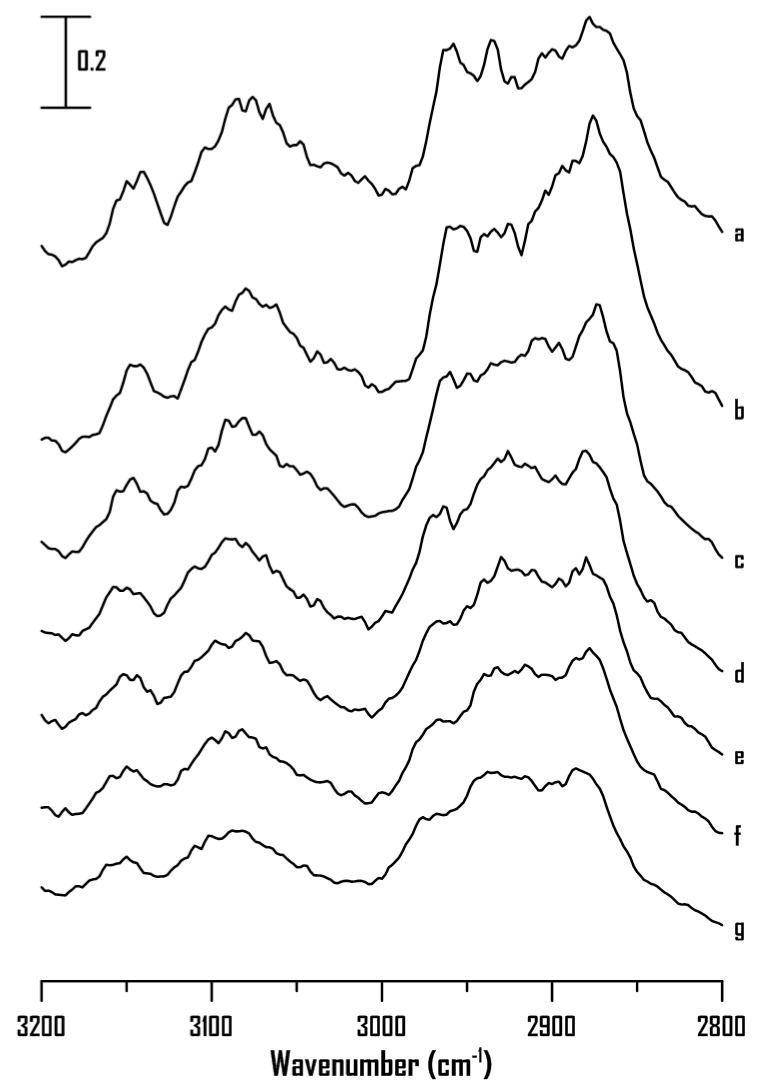
IR spectra of the [BMIM]Br-PEO mixture containing 80 wt% of [BMIM]Br at (**a**) ambient pressure and (**b**) 0.4, (**c**) 0.7, (**d**) 1.1, (**e**) 1.5, (**f**) 1.8, and (**g**) 2.5 GPa.

**Figure 5 ijms-22-00981-f005:**
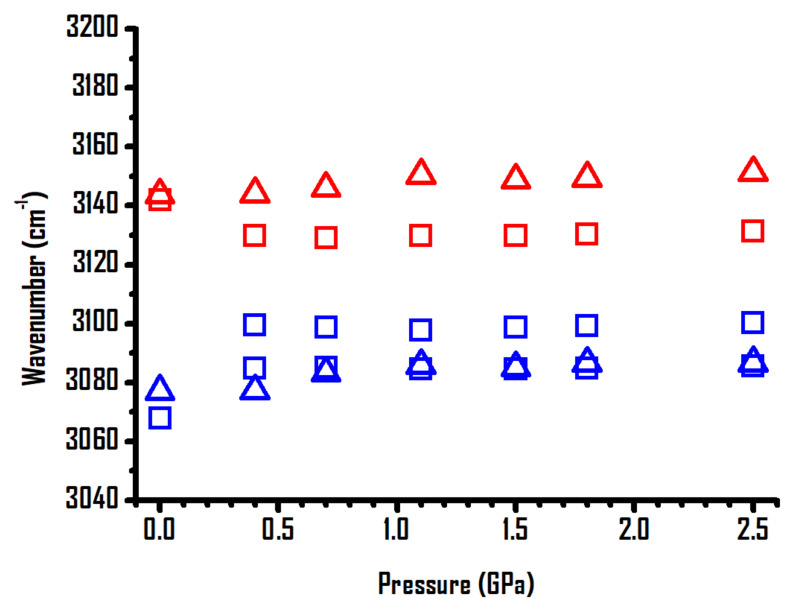
Pressure dependence of the C–H stretching frequencies of imidazolium C^4,5^–H and C^2^–H of pure [BMIM]Br (squares) and [BMIM]Br-PEO mixture containing 80 wt% [BMIM]Br (triangles).

**Figure 6 ijms-22-00981-f006:**
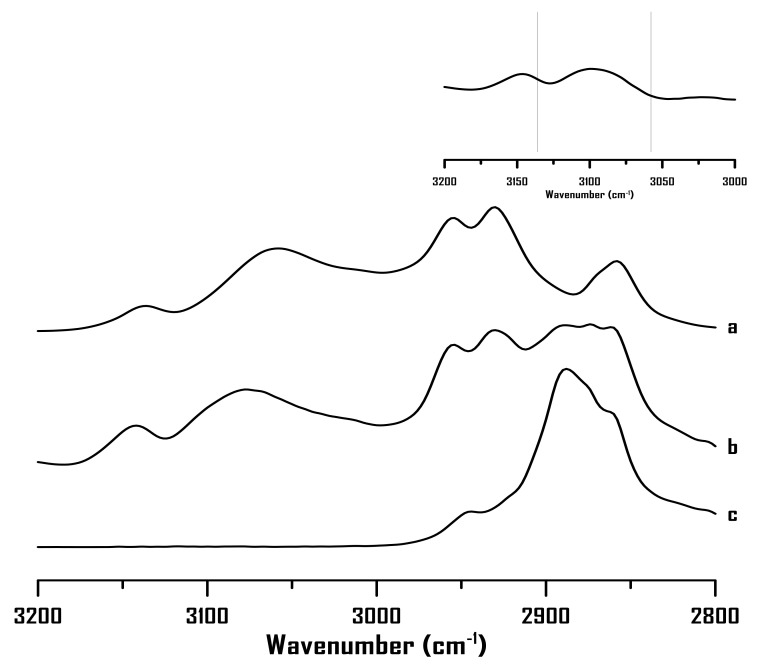
IR spectra of (**a**) pure [HMIM]Br, (**b**) mixture of PEO containing 75 wt% [HMIM]Br, and (**c**) pure PEO at ambient pressure. Upper-right corner: the difference spectrum (b-a).

**Figure 7 ijms-22-00981-f007:**
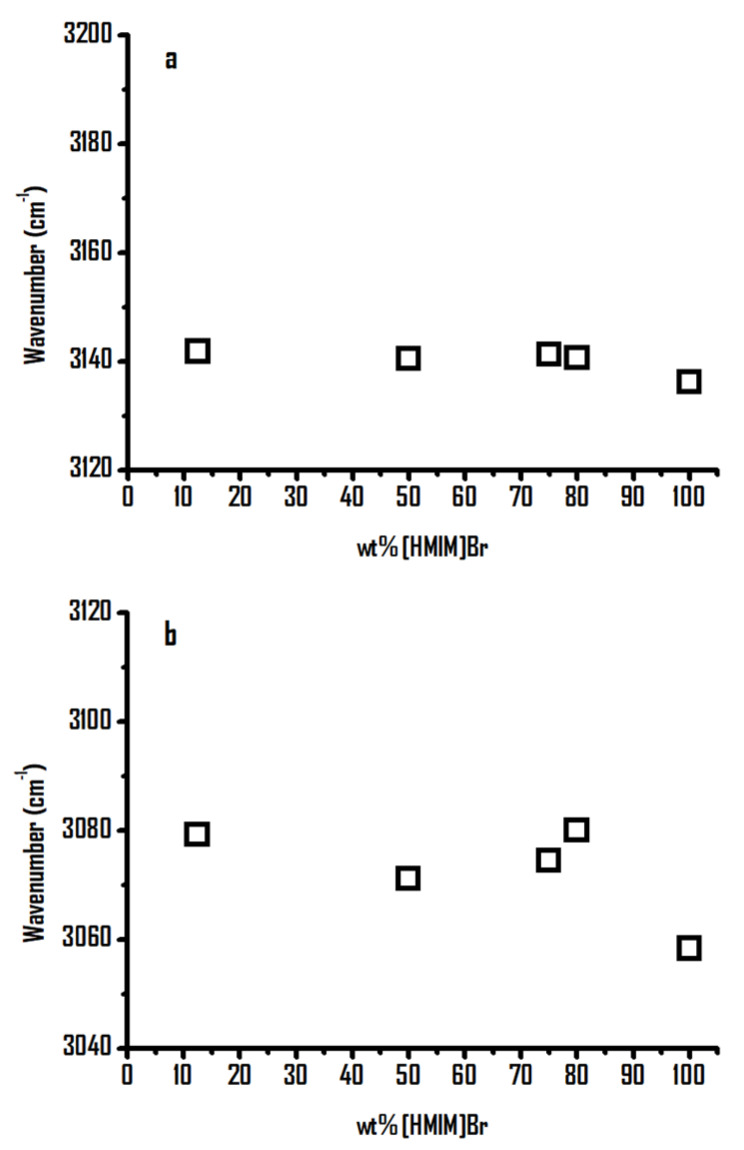
Concentration dependence of (**a**) imidazolium C^4,5^–H and (**b**) imidazolium C^2^–H stretching bands of [HMIM]Br-PEO mixtures as a function of the weight percentage of [HMIM]Br.

**Figure 8 ijms-22-00981-f008:**
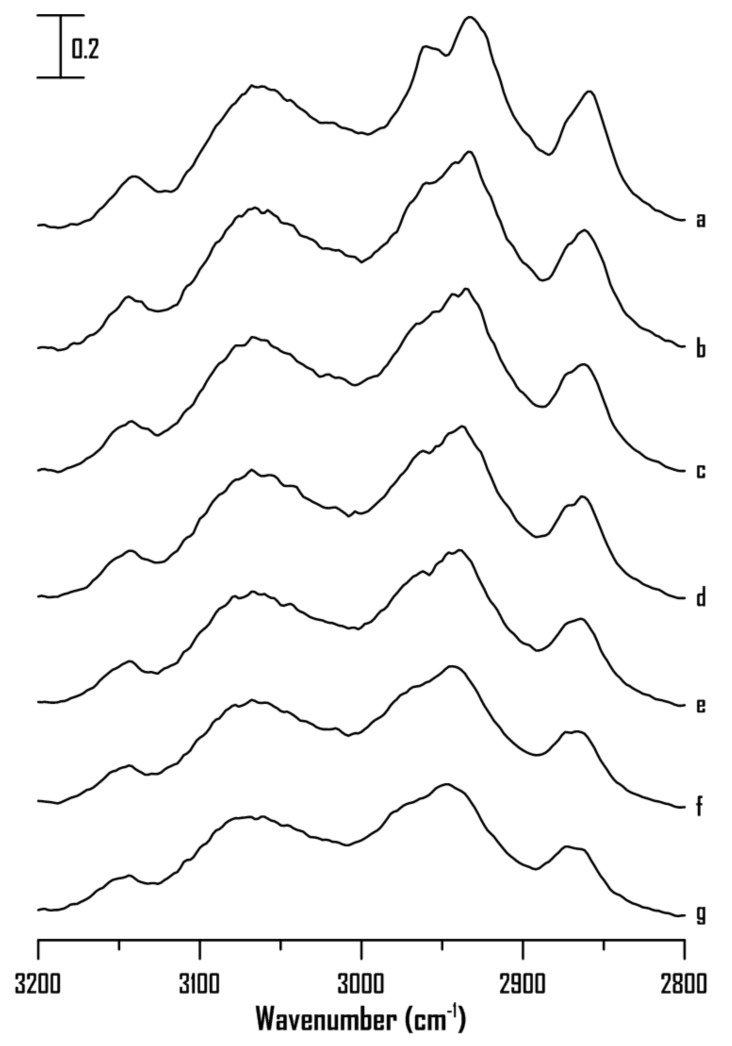
IR spectra of pure [HMIM]Br at (**a**) ambient pressure and (**b**) 0.4, (**c**) 0.7, (**d**) 1.1, (**e**) 1.5, (**f**) 1.8, and (**g**) 2.5 GPa.

**Figure 9 ijms-22-00981-f009:**
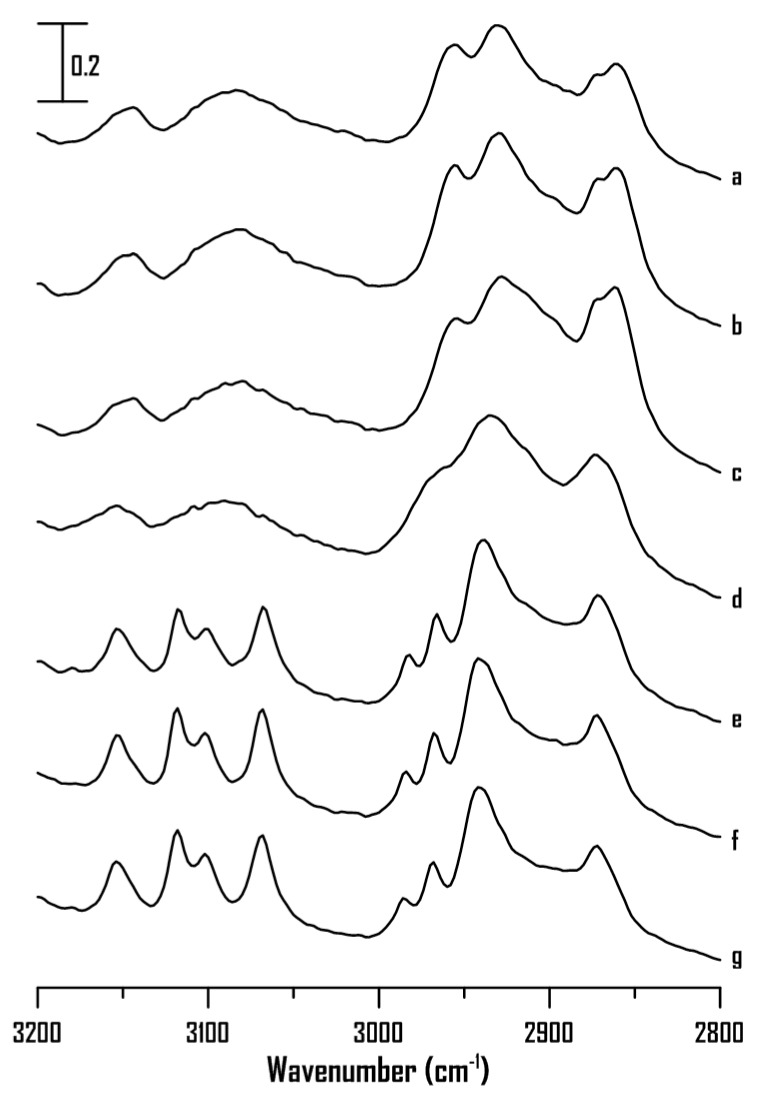
IR spectra of the [HMIM]Br-PEO mixture containing 80 wt% [HMIM]Br at (**a**) ambient pressure and (**b**) 0.4, (**c**) 0.7, (**d**) 1.1, (**e**) 1.5, (**f**) 1.8, and (**g**) 2.5 GPa.

**Figure 10 ijms-22-00981-f010:**
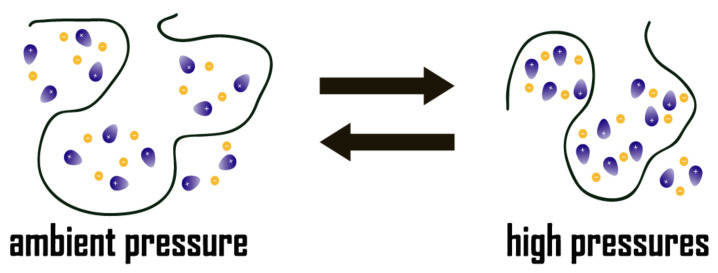
Illustration of the possible structural reorganization of [HMIM]Br-PEO mixtures.

## Data Availability

Data available on request. The data presented in this study are available on request from the corresponding author.

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
