# Peer review of "Structural Reorganization of Imidazolium Ionic Liquids Induced by Pressure-Enhanced Ionic Liquid—Polyethylene Oxide Interactions"

_ijms, 2021, doi:10.3390/ijms22020981_

Round 1

Reviewer 1 Report

This paper shoes an interesting result that alkyl methylimidazolium bromide displays a liquid structural transition induced by high pressure, from less ordered state to higher ordered state, by infrared spectroscopy.
The reviewer requests the authors some revision for readers’ understanding.

(1) The authors used some words indicating liquid structure states such as “cluster”, “network of IL”, “semi-crystalline network”, “semi-crystalline structure” and so on. Those words are not defined or explained well in this paper. Therefore, the readers might not well understand structural image for each words, then might not image what structural change occurs in the system. The reviewer requests the authors to add some explanations of those words to help the readers understand.

(2) Line 120
The narrowing and splitting peaks should mean higher ordered state. For achieving the higher ordering, is the word of “disturbance” is adequate? (Generally, disturbance bring about less ordering state, doesn’t it?) If the authors think that this word should be adequate, the reviewer recommends the authors to add some description for adequacy to help readers understand.

(3) Line 219
“the occurrence of ions on the long alkyl chain side”
What does it mean?

(4) Figure 10
If the words of “cluster” for ambient pressure and “network” for high pressures are considered, is the picture for each states image adequate?

Author Response

Reviewer 1:

Comments:

This paper shoes an interesting result that alkyl methylimidazolium bromide displays a liquid structural transition induced by high pressure, from less ordered state to higher ordered state, by infrared spectroscopy.
The reviewer requests the authors some revision for readers’ understanding.

(1)The authors used some words indicating liquid structure states such as “cluster”, “network of IL”, “semi-crystalline network”, “semi-crystalline structure” and so on. Those words are not defined or explained well in this paper. Therefore, the readers might not well understand structural image for each words, then might not image what structural change occurs in the system. The reviewer requests the authors to add some explanations of those words to help the readers understand.

Reply:

Agree.  Thank you for your good faiths to make our manuscript more readable. Based on the comments, the explanations and sentences were added.

In page 2 lines 49-53

“The presence of nano-segregation domains in ILs, i.e., heterogeneous microstructures, was reported in the literature[12]. The local nanoscale organization of imidazolium ILs can be described as supramolecular structures of ion pairs (cation-anion), clusters (cation-anion-cation-anion…), and/or three-dimensional network (cluster-cluster associations). ”

In page 11 lines 233-234

“[HMIM]Br can form a semi-crystalline structure (high-ordered 3-D structures of ions) in the presence of PEO ”

To be consistent, “semi-crystalline network” was replaced by “semi-crystalline structures” in the text.

(2) Line 120
The narrowing and splitting peaks should mean higher ordered state. For achieving the higher ordering, is the word of “disturbance” is adequate? (Generally, disturbance bring about less ordering state, doesn’t it?) If the authors think that this word should be adequate, the reviewer recommends the authors to add some description for adequacy to help readers understand.

Reply:

Agree.  Based on the suggestion of Reviewer 1, “local structure disturbance” was replaced by “local structure organization” in the text.

In page 5, lines 130-132, we wrote:

“The narrowing and splitting of the imidazolium C–H bands may indicate the local structure organization of cations via pressure-enhanced interactions, as shown in Figure 3b. ”

(3)Line 219
“the occurrence of ions on the long alkyl chain side”
What does it mean?

Reply: 

Agree to change. Please accept my apology for the vague sentence.  We rewrote:

 In page 11, lines 237-239

“Researchers have revealed that the long alkyl chain disrupts the symmetry of each side of the imidazolium ring, and the presences of anions on the side of the long alkyl C-H chain is less favored[4].  ”

(4)Figure 10
If the words of “cluster” for ambient pressure and “network” for high pressures are considered, is the picture for each states image adequate?

Reply:

Agree. As suggested by Reviewer 1, Figure 10 is replaced by a new Figure in Page 13 to reflect “cluster” for ambient pressure and “network” for high pressures.  

Reviewer 2 Report

This is an interesting paper, which potentially could have interesting applications. However, the presented results are too descriptive, and they might not be directly applied to other systems. The manuscript would also benefit from additional information, discussion, and potentially experiments.

There are several major concerns that should be addressed:

  1. Although the choice of PEO is clear and justifiable, a) wouldn’t other glycols also cause similar effects? and b) how would PEO with other MWs affect the organization the ionic liquids (i.e., why only one PEO was considered)? Based on the observed changes, it might be argued that glycols or polyols might exhibit the similar effects. Some discussion is warranted.
  2. In addition, some control experiment(s) is (are) warranted. For example, either ethylene glycol vs PEO, or maybe even a polymer that does not possess the ability to disrupt the aggregates nor has H-bonding possibilities.
  3. Aggregates/clusters of ionic liquids are likely to be similar for the chosen ionic liquids, simply because a two methylene unit difference is unlikely to cause drastic structural alterations.
  4. The choice of ionic liquids, e.g., [C4-mim]Br and [C6-mim]Br , needs some additional justification. C4- and C6- are fairly similar, and therefore the physical properties, including the aggregate behavior, are also expected to be similar. The pressure-dependent results (Figures 3 and 8) show that the changes are fairly minor. Despite the fact that the IR spectra of ionic liquids (neat and PEO-mixtures) look different, the changes that are occurring appear to be fairly similar, and relatively insignificant.
  5. What is the evidence of the semi-crystalline nature of the ionic liquid – PEO mixtures? Some pictures and/or more elaborate discussion based on literature evidence should be provided. In addition, it’s stated that such networks are pressure induced, how is that confirmed from the presented IR spectra? In other words, what are the specific spectral changes that lead to this conclusion?
  6. Description of the spectral changes is a bit simplistic. The manuscript would significantly benefit from more detailed information; specifically, how and why sharpening/shifting of the peaks are related to structural organization. It is particularly clear, why the observed changes should refer to the re-organization of the cluster rather than to simply reflect the closer contacts between the ionic liquid clusters as the pressure increases?

There are a few minor issues:

  1. Figure 5 and Figure S3: the color labeling, which differentiates between H4/5 and H2, will enhance the visual readability of the results.
  2. Line 46: this appears to be poorly worded sentence (cations and anions cannot be crystalline; but the combination of asymmetric cations and anions might produce crystalline materials), which is based on one of the earlier reviews/accounts on ionic liquids. There are ample of examples of ionic liquids with symmetric anions and cations that possess phase transition temperatures below room temperature.
  3. Line 94: H2 is known to be most prone to the effect of the H-bonding (e.g. based on ample 1H NMR data). Thus, some correlations with known spectroscopic technique (even though are typically done under ambient pressure) should be made as opposed to speculations.
  4. It might be interesting to see overlayed spectra rather stacked ones, to see the differences in the IR-spectra. Further, difference spectra might/should be considered as well. These spectral manipulations should be more indicative of the specific changes or lack thereof.
  5. Line 250: the “shuttling of aggregation states” appears to be a speculation at this time.
  6. Although Figure 10 is a speculation, the graphic representation is a bit misleading. Regardless of the pressure, aggregates and some organized clusters must be present. Thus, there is some reorganization of those clusters/aggregates, but based on the observed changes in IR spectra, the structural differences between those states cannot be significant. However, the figure should unordered and ordered assemblies; thus, it might be warranted to redraw the figure.

Author Response

Comments:

This is an interesting paper, which potentially could have interesting applications. However, the presented results are too descriptive, and they might not be directly applied to other systems. The manuscript would also benefit from additional information, discussion, and potentially experiments.

There are several major concerns that should be addressed:

  1. Although the choice of PEO is clear and justifiable, a) wouldn’t other glycols also cause similar effects? and b) how would PEO with other MWs affect the organization the ionic liquids (i.e., why only one PEO was considered)? Based on the observed changes, it might be argued that glycols or polyols might exhibit the similar effects. Some discussion is warranted.

Reply:

Agree.  I think I learned a lot from this helpful comment. I feel I have better understanding after performing further experiments suggested by Reviewer 2.  Polyethylene glycol 1500 (PEO with low M.W.)-[HMIM]Br mixture were studied, and results was displayed in Figure S4 (in the Supplementary Materials). As shown in Figure S4, the band-narrowing of C–H absorptions is not detected for the [HMIM]Br-PEG 1500 (PEO with low M.W.) mixture containing 80 wt% of [HMIM]Br under high pressures.  We would like to emphasize that the band-narrowing of [HMIM]Br induced by PEO (M.W.~900,000) under high pressures is quite unique (or anomalous).  We have reported several similar systems including ILs-PVdF-co-HEP (Nanomaterials 2020, 10, 1973), ILs-DNA (Materials 2019, 12, 4202), and ILs-porous silica (Nanomaterials 2019, 9, 620), and such sharpening in IL absorptions induced by polymers under high pressures has not been observed before.  We added the following sentences:

In page 13, lines 292-301

“To check the effect of PEO with other molecular weights (M.W.) on the organization of [HMIM]Br, the pressure dependence of  [HMIM]Br -polyethylene glycol 1500 (PEG 1500 ,i.e., PEO with low M.W.) mixture and [HMIM]Br -ethylene glycol (EG) mixture were studied and displayed in Figure S4 and Figure S5, respectively, in the Supplementary Materials. As shown in Figure S4 and S5, the band-narrowing of C–H absorptions is not detected for the [HMIM]Br-PEG 1500 (PEO with low M.W.) and [HMIM]Br-EG mixtures containing 80 wt% of [HMIM]Br under high pressures.  In other words, the organization of [HMIM]Br may be sensitive the molecular weight of polyethylene oxide under high pressures. We note that the [HMIM]Br-PEG 1500 and [HMIM]Br-EG mixtures are in the state of liquid under ambient pressure. Nevertheless, [HMIM]Br-PEO (MW. 900,000) mixtures are gel-like under ambient pressure.”   

  1. In addition, some control experiment(s) is (are) warranted. For example, either ethylene glycol vs PEO, or maybe even a polymer that does not possess the ability to disrupt the aggregates nor has H-bonding possibilities.

Reply:

Agree.  Thank you for your precious time. Based on the helpful comments from Reviewer 2, Ethylene glycol (EG)-[HMIM]Br mixture were studied, and results was displayed in Figure S5 (in the Supplementary Materials). The band-narrowing of C–H absorptions is not detected for the [HMIM]Br-EG mixture containing 80 wt% of [HMIM]Br under high pressures.

We added:

In Page 14, lines 350-352

“Figure S5. IR spectra of the [HMIM]Br-ethylene glycol mixture containing 80 wt% [HMIM]Br at (a) ambient pressure and (b) 0.4, (c) 0.7, (d) 1.1, (e) 1.5, (f) 1.8, and (g) 2.5 GPa. ”

  1. Aggregates/clusters of ionic liquids are likely to be similar for the chosen ionic liquids, simply because a two methylene unit difference is unlikely to cause drastic structural alterations.

Reply:

Agree.  We agree that aggregates/clusters of [C4-mim]Br and [C6-mim]Br are likely to be similar under ambient pressure.  However, high pressure measurements demonstrate that [C4-mim]Br and [C6-mim]Br can be distinguished under high pressures.  We added the following

In page 11, lines 254-264

“It is known that aggregates/clusters of [HMIM]Br and [BMIM]Br are likely to be similar under ambient pressure.  The almost identical trends of concentration-dependence observed in Figure 2 ([BMIM]Br-PEO) and Figure 7 ([HMIM]Br-PEO) suggest that IR measurements obtained under ambient pressure are not a sensitive tool to distinguish the local structures between [BMIM]Br-PEO and [HMIM]Br-PEO. This study indicates that high pressures can serve as a useful method to probe the differences in [BMIM]Br-PEO and [HMIM]Br-PEO interactions along the surface of PEO matrix under high pressures.  Previous studies also revealed that the associated structures of [C4MIM][PF6]/DNA are more stable than those of [C3MIM][PF6]/DNA under high pressures[30].  ”    

  1. The choice of ionic liquids, e.g., [C4-mim]Br and [C6-mim]Br , needs some additional justification. C4- and C6- are fairly similar, and therefore the physical properties, including the aggregate behavior, are also expected to be similar. The pressure-dependent results (Figures 3 and 8) show that the changes are fairly minor. Despite the fact that the IR spectra of ionic liquids (neat and PEO-mixtures) look different, the changes that are occurring appear to be fairly similar, and relatively insignificant.

Reply:

  Partly agree. We agree that the pressure-dependent results (Figures 3 and 8) show that the changes are fairly minor.  However, the sharpening and splitting of C2-H band (~ 3080 cm-1) at 0.4 GPa (Figure 3b) is observable. We describe this observation in Page 5, lines 130-132

“The narrowing and splitting of the imidazolium C–H bands may indicate the local structure organization of cations via pressure-enhanced interactions, as shown in Figure 3b.”

  1. What is the evidence of the semi-crystalline nature of the ionic liquid – PEO mixtures? Some pictures and/or more elaborate discussion based on literature evidence should be provided. In addition, it’s stated that such networks are pressure induced, how is that confirmed from the presented IR spectra? In other words, what are the specific spectral changes that lead to this conclusion?

Reply:

  Agree. As suggested by Reviewer 2, we agree that more discussion is needed.  The sentences were added

In page 11, lines 224-230

“The band narrowing observed in Figure 9e-g may be related to the decrease in inhomogeneous broadening. The sharper structures revealed in Figure 9e-g is in part due to conformation drift and anisotropic environment in a solid structures.  For example, water is transformed from liquid to high-pressure ice VI and VII at the pressure of ~1 and ~2 GPa, respectively[39].  The reduction of O-H bandwidths, being obvious for high-pressure ice VI and VII, arises from the change of geometric properties of the hydrogen-bond network[39]. ”

  1. Description of the spectral changes is a bit simplistic. The manuscript would significantly benefit from more detailed information; specifically, how and why sharpening/shifting of the peaks are related to structural organization. It is particularly clear, why the observed changes should refer to the re-organization of the cluster rather than to simply reflect the closer contacts between the ionic liquid clusters as the pressure increases?

Reply:

Agree. Based on the comments of Reviewer 2, the following sentences were added.

In page 11, line 249-264

“Pressures used in this study (0.1 MPa – 2.5 GPa) mainly change intermolecular (or interionic) distances (closer contact) and affect conformations[40] .  Actually in order to change the electronic structure of molecules or ions, pressure > 30 GPa are required.  The abrupt changes (or dicontinuities) in frequency-shifts and bandwidths upon compression may provide the clues for conformation transitions or pressure-induced phase transitions.  For example, pressure-induced reversible unfolding of biomolecules has drawn the attention of researchers[40]. It is known that aggregates/clusters of [HMIM]Br and [BMIM]Br are likely to be similar under ambient pressure.  The almost identical trends of concentration-dependence observed in Figure 2 ([BMIM]Br-PEO) and Figure 7 ([HMIM]Br-PEO) suggest that IR measurements obtained under ambient pressure are not a sensitive tool to distinguish the local structures between [BMIM]Br-PEO and [HMIM]Br-PEO. This study indicates that high pressures can serve as a useful method to probe the differences in [BMIM]Br-PEO and [HMIM]Br-PEO interactions along the surface of PEO matrix under high pressures.  Previous studies also revealed that the associated structures of [C4MIM][PF6]/DNA are more stable than those of [C3MIM][PF6]/DNA under high pressures[30]. ”

There are a few minor issues:

  1. Figure 5 and Figure S3: the color labeling, which differentiates between H4/5 and H2, will enhance the visual readability of the results.

Reply:

Agree.   Based on the comments of Reviewer 2, the color labeling was added to enhance the readability.

  1. Line 46: this appears to be poorly worded sentence (cations and anions cannot be crystalline; but the combination of asymmetric cations and anions might produce crystalline materials), which is based on one of the earlier reviews/accounts on ionic liquids. There are ample of examples of ionic liquids with symmetric anions and cations that possess phase transition temperatures below room temperature.

Reply:

Agree.  Please accept my apology for poor writing.  Based on the comments, we rewrote:

In Page 2, lines 45- 47

“The cations and anions of the imidazolium-ILs attract each other, but ions in ILs do not undergo ready packing. Thus, the liquid phase of ILs is typically observed at room temperature. ”    

  1. Line 94: H2 is known to be most prone to the effect of the H-bonding (e.g. based on ample 1H NMR data). Thus, some correlations with known spectroscopic technique (even though are typically done under ambient pressure) should be made as opposed to speculations.

Reply:

Agree.  We tried to mention the results of 1H NMR and MD simulations in this revised manuscript, based on the helpful review article (Ref.12).  We added:

In page 3, lines 103-105

“Based on the results of 1H NMR and MD simulations [12], C2-H is most prone to the effects of the hydrogen bonding.  For example, the relative distance of anion is close to C2-H and water for [BMIM][BF4]-water mixtures[12]. ”

  1. It might be interesting to see overlayed spectra rather stacked ones, to see the differences in the IR-spectra. Further, difference spectra might/should be considered as well. These spectral manipulations should be more indicative of the specific changes or lack thereof.

Reply:

Agree. Please accept the apology for my old pattern of reading.  Frankly speaking, I have the difficulty to read the overlaid spectra (probably because I am a little bit old).  I usually print the papers out to read (old style) instead of reading them in computer (new style), and the colors in overlaid spectra disappear for me.  Therefore stacked spectra were chosen by me.  For the convenience of comparison and the new style readers, the overlaid spectra are prepared and displayed in Figure S 6-Figure S11.  We added

In Page 13, lines 301-303

“By the way, overlaid spectra (Figures 1, 3, 4, 6, 8, 9) are displayed in Figure S6- Figure S11, respectively, for the purpose of comparison.  ”     

  1. Line 250: the “shuttling of aggregation states” appears to be a speculation at this time.

Reply:

   Agree. I tried to rewrote the sentence as follows.

In page 13, lines 286-288

“The differences in aggregation states for ambient-pressure phase and high-pressure phase may suggest the potential of [HMIM]Br-PEO (M.W~900,000) for serving as optical or electronic switches ”  

  1. Although Figure 10 is a speculation, the graphic representation is a bit misleading. Regardless of the pressure, aggregates and some organized clusters must be present. Thus, there is some reorganization of those clusters/aggregates, but based on the observed changes in IR spectra, the structural differences between those states cannot be significant. However, the figure should unordered and ordered assemblies; thus, it might be warranted to redraw the figure.

Reply:

Agree.  Figure 10 is replaced by a new figure to reflect some organization under ambient pressure and a little bit higher ordered organization under high pressure.

Round 2

Reviewer 2 Report

The authors addressed all the comments, and the manuscript could be considered for publication.